# The Effect of Different g-C$_3$N$_4$ Precursor Nature on Its Structural Control and Photocatalytic Degradation Activity

Xiuhang Liu [1,†], Xiaoye Xu [1,†], Huihui Gan [1,2,*], Mengfei Yu [1] and Ying Huang [1,*]

1 School of Civil and Environmental Engineering, Ningbo University, Ningbo 315211, China
2 Institute of Ocean Engineering, Ningbo University, Ningbo 315211, China
* Correspondence: ganhuihui@nbu.edu.cn (H.G.); huangying1@nbu.edu.cn (Y.H.)
† These authors have contributed equally to this work and share first authorship.

**Abstract:** Due to its good visible-light photocatalytic activity and environmental friendliness, g-C$_3$N$_4$ has attracted much attention. The relationship between precursor type and the properties of obtained catalysts is interesting to investigate. In this work, target catalysts were prepared via the thermal polymerization of different precursors, melamine, a mixture of urea and melamine, and a mixture of melamine and cyanuric acid. The prepared g-C$_3$N$_4$ were characterized by X-ray diffraction (XRD), Fourier transform infrared spectrometry (FT-IR), UV–vis diffuse reflectance spectroscopy (UV–vis DRS), and scanning electron microscopy (SEM). Through the characterization and analysis, the adjusting of precursors could result in the change of the microstructure. The maximum BET surface area was 98.04 cm$^3$g$^{-1}$ through precursor controlling, more than eight times that of MCN (11.15 cm$^3$g$^{-1}$) using melamine as precursor. The thermal decomposition process was also analyzed to discuss the interaction and polymerization with precursor controlling. The introduction of melamine and cyanuric acid with melamine as precursors led to the formation of a special nanotube structure and additional function groups on the surface of g-C$_3$N$_4$ to increase the photocatalytic activity.

**Keywords:** g-C$_3$N$_4$; photocatalytic activity; precursors; microstructure; antibiotics

## 1. Introduction

Antibiotics, as an antimicrobial drug and growth promoter, are widely used in human clinical and veterinary drugs. Since the outbreak of COVID-19, human dependence on antibiotics has expanded, and the abuse of antibiotics has become more and more serious [1]. Various antibiotics have been frequently detected in natural bodies of water [2,3], which is a serious problem for the ecological environment and human health.

In recent years, the advanced oxidation process (AOPs) has attracted significant attention because of its fast reaction rate, efficient high-concentration wastewater treatment, and high degree of mineralization. AOPs was proposed to produce highly oxidized hydroxyl radicals to mineralize pollutants. This technology has also developed a variety of processes, which can be distinguished according to different oxidants, such as ozone oxidation with ozone (O$_3$) [4], Fenton oxidation with hydrogen peroxide (H$_2$O$_2$) [5], and persulfate with persulfate [6]. Photocatalytic oxidation technology can directly use light energy to produce oxidation-free radicals. Compared with other methods, it possesses the advantages of being more green, efficient, and low cost, showing a good application prospect in antibiotic wastewater treatment. Preparing high efficiency photocatalysts or improving photocatalysts through specific technology is essential for photocatalytic technology, which is also the focus of current research.

Carbon nitride (C$_3$N$_4$) as a metal-free polymeric material has attracted much attention and has been considered a promising photocatalyst because of its unique electronic properties [7–10]. There are several allotropes for C$_3$N$_4$, in which the graphitic carbon nitride (g-C$_3$N$_4$)) is regarded as the most stable form and most intensively studied [11].

In recent years, g-C$_3$N$_4$ has increasingly developed in many fields, such as water splitting [12], heavy metal adsorption [13], pollutant degradation [14], and bacteria disinfection [15]. Since g-C$_3$N$_4$ was widely concerned, researchers at home and abroad began to invest a lot of energy and try various means to synthesize this new material.

Up to now, the commonly used preparation methods include thermal polymerization, supramolecular self-assembly, solvent thermal synthesis, high temperature and high-pressure synthesis, electrochemical deposition, and physical/chemical vapor deposition. Wang et al. [16] synthesized a O-doped g-C$_3$N$_4$ catalyst by the solvothermal method at a relatively low temperature. By this template-free way, they obtained hollow microspheres of O-doped g-C$_3$N$_4$, which showed good photocatalytic activity for the pollutant degradation and water hydrogen decomposition. Chebanenko et al. [17] deposited g-C$_3$N$_4$-based composites by oxide process vapor deposition (OPVD), which can be used as a multi-junction carbon photocatalyst owing to a wider working spectrum. Wang et al. [18], using melamine as a precursor and three-dimensional cubic mesoporous silica KIT-6 as a template, prepared ordered cubic mesoporous g-C$_3$N$_4$ by simple chemical vapor deposition. The obtained ordered mesoporous g-C$_3$N$_4$ showed higher photocatalytic activity than sheet g-C$_3$N$_4$ in the reduction of CO$_2$ and H$_2$O. Liu et al. [19] prepared C-doped hollow spherical g-C$_3$N$_4$ derived from supramolecular self-assembly of melamine, glucose, and cyanuric acid, which exhibited enhanced photoredox water-splitting with a hydrogen yield of 305 mmol h$^{-1}$, 28.5 times than that of bulk g-C$_3$N$_4$.

Among various synthesis methods of g-C$_3$N$_4$, thermal polymerization is a facile method with no additive except the precursor itself. The N-rich molecular precursors such as melamine, urea, and thiourea have been widely used for photocatalysis. However, the g-C$_3$N$_4$ obtained from polycondensation reactions of melamine has many limits, such as the low specific surface area and a high charge carrier recombination rate under solar-light excitation, resulting in low photocatalytic activity.

Recently, there are many attempts at improving the photocatalytic performance of g-C$_3$N$_4$, such as doping metal elements, coupling with other semiconductors, and nanostructure engineering. Element doping can adjust the valence band and conduction band potential of materials, thus improving the photocatalytic performance. P/Cl co-doped carbon nitride nanosheets with nitrogen defects were prepared by one-step heat treatment [20]. P and Cl atoms successfully entered the structural units of g-C$_3$N$_4$, creating more defects simultaneously. These newly generated defects can significantly improve the photocatalytic performance of g-C$_3$N$_4$. The prepared materials show more efficient and stable photocatalytic activity for the photocatalytic decomposition of RhB and NOR under visible light irradiation. Chang et al. [21] prepared Pd-doped mesoporous g-C$_3$N$_4$ and used it to degrade bisphenol A in water. The surface doping with Pd can improve the optical absorption of mesoporous g-C$_3$N$_4$ in the UV-vis range. When it comes to material coupling, it is regarded as a practical approach to promote charge separation of photoexcited electron–hole pairs. Coupling with other semiconductors is a good approach to forming a highly efficient transport path for the photogenerated electron-hole pairs. Cao et al. [22] reported that the g-C$_3$N$_4$/WO$_3$ nanoplate composite, showed as a direct solid-state Z-scheme heterojunction, drastically improved the photocatalytic efficiency for NO removal, which was approximately 2.1 times higher than that of pure g-C$_3$N$_4$. Tang et al. [23] successfully synthesized TiO$_2$/g-C$_3$N$_4$ composite nanofiber materials by electrospinning with titanium n-butyl ether and urea. Under visible-light irradiation, the degradation rate of rhodamine B by the composite was 6.4 times higher than that of pure g-C$_3$N$_4$.

Besides the mentioned modifying strategies, the microstructure fabrication of g-C$_3$N$_4$ has been an effective method to boost its photocatalytic performance. The porous structure in morphology regulation has a higher surface area, porosity, and carrier separation and mobility than the block g-C$_3$N$_4$, thus improving its photocatalytic activity. Liang et al. [24] obtained ultra-thin g-C$_3$N$_4$ nanosheets with large carbon vacancies and higher specific surface area by heat-treating bulk g-C$_3$N$_4$ under NH$_3$, and the transfer and diffusion rate of photoelectron-hole pairs were significantly improved. Self-templating of melamine with

urea, cyanuric acid, and cyanuric chloride, is a simple, safe, and cheap method to tune porous and hollow microstructures, such as nanorods [25], nanosheets [26], nanoflower [27], and nanotubes [28] of g-$C_3N_4$. In addition, the aggregates can be generated from melamine and other N-rich molecular precursors through noncovalent interactions, such as hydrogen bonding, to form well-organized g-$C_3N_4$ precursors. Therefore, it is significant to explore the influence of different precursors on the microstructure and photocatalytic activity of g-$C_3N_4$ to understand further the microstructure of carbon nitride and its influence on photocatalytic efficiency.

The photocatalytic activity of the prepared catalyst was evaluated by the decomposition of norfloxacin (NOR) antibiotics in solution. NOR, as a quinolone antibiotic, is widely used in medical and animal husbandry fields. However, NOR usually has an incomplete metabolism in organisms; 60–90% in the form of original agents or metabolites are excreted into the environment, and with the natural water circulation migration to surface water, groundwater, and even drinking water, causing serious harm to the aquatic ecosystem and human health [29–31]. Nowadays, reducing the release of antibiotics into aquatic environments is an urgent concern. In addition, antibiotics were proven to be particularly resistant to wastewater treatment.

In this work, g-$C_3N_4$ was synthesized by the thermal decomposition of melamine, a mixture of urea and melamine, and a mixture of melamine and cyanuric acid as precursors, respectively. As a method to study the thermodynamic behavior of materials, thermal decomposition kinetics is expected to solve the relationship between precursor structure and target catalyst structure. The thermal decomposition kinetics were studied using thermogravimetry (TG) to further analyze the differences in thermal polymerization by using melamine as a single precursor and combined precursors for the preparation of g-$C_3N_4$. The changing micro-morphology by controlling precursors was also discussed. This paper aims to reveal the effect of adjusting precursors on the microstructure of target products and visible-light photocatalytic activity for the degradation of NOR in solution.

## 2. Results and Discussion

### 2.1. Structure and Morphology of g-$C_3N_4$ Prepared by Different Precursors

The crystal structure of the prepared g-$C_3N_4$ samples including MCN, UMCN, and CMCN was studied by XRD. The thermal decomposition of melamine synthesized the catalyst samples, denoted as MCN. The obtained samples using a mixture of urea and melamine or a combination of melamine and cyanuric acid as precursors were denoted as UMCN and CMCN, respectively. As shown in Figure 1, the characteristic peaks of all g-$C_3N_4$ samples were similar. The sharp and strong peak at about 27.6° assigned to the (002) facet of g-$C_3N_4$ was associated with the stacking of the conjugated aromatic sheets. Another characteristic peak at 13.1° assigned to the (100) facet was ascribed to in-plane nitrogen linkages of tri-s-triazine motifs. In addition, the peak intensity of UMCN was dramatically lower than that of MCN and CMCN, implying the decease of crystal growth and a looser structure through the involvement of urea. As previously reported [32], during the decomposition processes of the urea, a large amount of $NH_3$ formed. This small and polar molecule could intercalate into the interplanar space and break the weak van der Waals bonds of the product. Thus, through adding urea to the mixed precursors, it was more accessible to obtain the smaller-sized nanoparticle products. When it comes to the CMCN, the diffraction peak at 27.6° shifted to 27.4° compared with MCN, indicating that with the presence of cyanuric acid, the lattice plane distance was increased.

The molecular structure of MCN, UMCN, and CMCN were characterized by Fourier transform infrared (FTIR) spectrometry, and the FTIR spectra are shown in Figure 2. The similar absorption peaks of all samples further confirmed the basic molecular composition of g-$C_3N_4$. The strong absorption peak at 815 $cm^{-1}$ corresponds to the vibration of the triazine units, which could be formed through the polymerization of cyanic acid in the gaseous phase [33]. The absorption bands within the 1100–1700 $cm^{-1}$ region correspond to C–N and C=N stretching modes of N-containing heterocycles. The absorption peaks

between 3000–3500 cm$^{-1}$ were assigned to the stretching vibration of uncondensed terminal amino N–H bonds. The spectral bands around 3500 cm$^{-1}$ correspond to the stretching vibrations of -OH groups. It is noted that the intensity of the absorption peak of UMCN between 3000–3500 cm$^{-1}$ corresponding to the related amino groups was more enhanced than that of the MCN, which implies that a large number of amino groups were introduced onto the surface of the g-C$_3$N$_4$ during the thermal polymerization process. As for CMCN, the broadened absorption peak between 3000–3500 cm$^{-1}$ implies the relatively lower content of the N–H groups and probably introduced more -OH groups or oxygen containing groups.

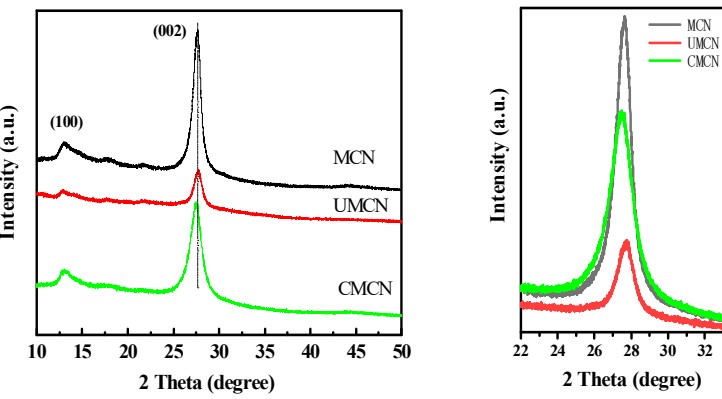

**Figure 1.** XRD patterns of the samples MCN, UMCN, and CMCN.

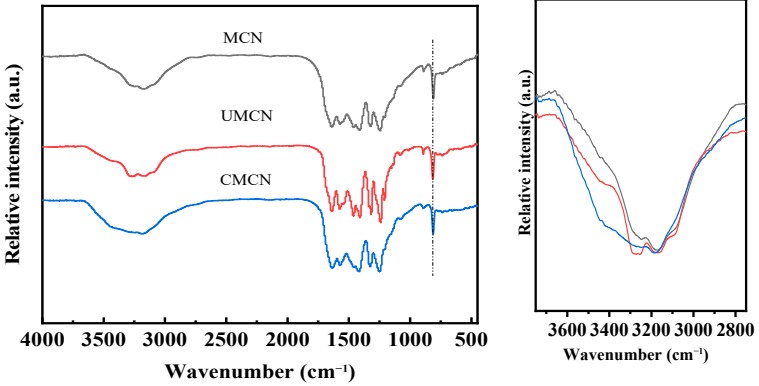

**Figure 2.** FT-IR spectra of the samples MCN, UMCN, and CMCN.

The morphology and microstructure of MCN, UMCN, and CMCN were investigated by SEM. Figure 3a illustrates that the MCN consisted of a large number of blocky structures, which had some folds on it. As shown in Figure 3b–d, both UMCN and CMCN displayed one-dimensional tubular structures. The UMCN nanotubes had a diameter of 200 nm, while the CMCN nanotubes had a diameter around 100 nm. The one-dimensional tubular structure was formed because urea was first converted into cyanuric acid in situ during calcination, and further formed cyanamine-cyanuric acid supramolecular polymer nanorods through hydrogen bonding with melamine. The similar nanorods were formed when a mixture of melamine and cyanuric acid was heated [34]. One-dimensional nanotubes can enhance the absorption of incident light and promote carrier migration along 1D length, thus promoting effective carrier separation. The exposed inner and outer surfaces of nanotubes had larger specific surface areas, thus providing more catalytic active sites [35]. Figure 3c shows that a large number of holes were distributed on the CMCN nanotubes, which were formed due to the release of gases during the formation of g-C$_3$N$_4$ nanotubes [36]. Scheme 1 illustrates the nanotube microstructure formation of g-C$_3$N$_4$ through one-pot thermal decomposition with different precursor controlling. In addition,

porous g-C$_3$N$_4$ had better photocatalytic properties due to its larger surface area and many channels that facilitated mass diffusion [37]. There are many reported porous structures and the preparation is summarized in Table 1.

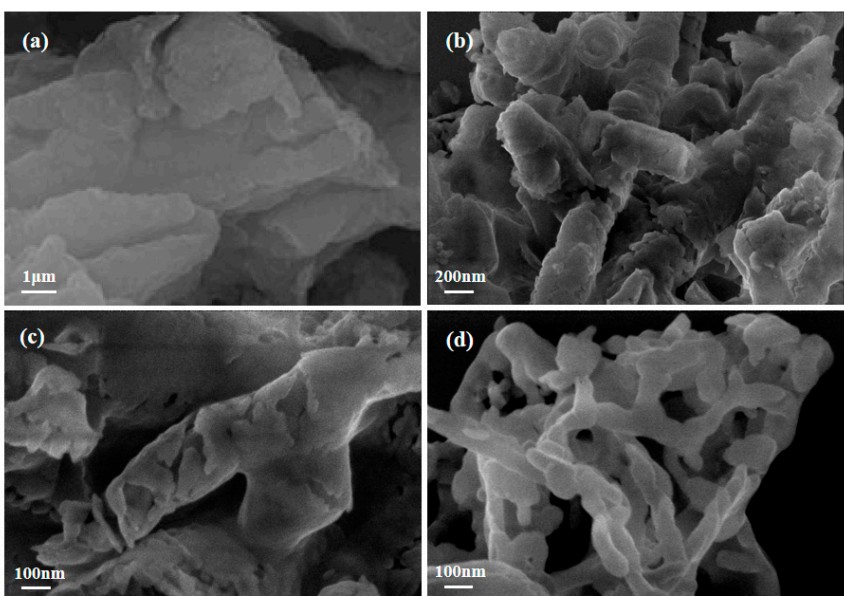

**Figure 3.** SEM images of MCN (**a**), UMCN (**b,c**), and CMCN (**d**).

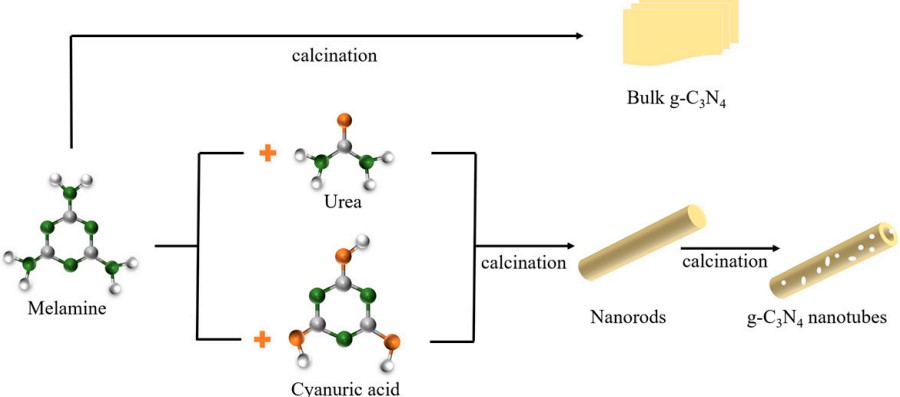

**Scheme 1.** The formation of g-C$_3$N$_4$ nanotubes by one-pot thermal decomposition.

**Table 1.** Summary of reported porous structure and the preparation method.

| Porous Structure | Precursor | Method | Ref. |
|---|---|---|---|
| Short rod-like | Dicyandiamide with SBA-15 template | High pressure and High temperature (HPHT) process | [38] |
| Nanosheets | Melamine and ammonium bicarbonate | Solid-state reaction | [39] |
| Crafted 1D mesoporous microtubes | Dicyandiamide | Liquid-liquid interfacial self-assembly strategy | [40] |
| Nanotube | Melamine and urea | Thermal polymerization | This work |
| Nanotube | Melamine and cyanuric acid | Thermal polymerization | This work |

The BET surface area, pore volume, and average pore size of MCN, CMCN, and UMCN are summarized in Table 2. The BET surface area of UMCN is 98.04 m$^2$/g, which is

more than 8 times more than that of CMCN and MCN. Meanwhile, the total pore volume of MCN is 0.10 cm$^3$/g, while it increased to 0.58 cm$^3$/g for UMCN. The results imply that the UMCN could possess the best adsorption ability and more active sites for the photocatalytic degradation.

**Table 2.** BET surface area, pore volume, and average pore size of MCN, CMCN, and UMCN.

| Samples | MCN | CMCN | UMCN |
|---|---|---|---|
| BET surface area (m$^2$/g) | 11.15 | 11.87 | 98.04 |
| Total pore volume (cm$^3$/g) | 0.10 | 0.12 | 0.58 |
| Average pore size (nm) | 36.31 | 40.01 | 23.68 |

The photoabsorption ability was analyzed using UV–vis diffused reflectance spectroscopy. As shown in Figure 4a, the UMCN and CMCN displayed a distinct blue shift of the intrinsic absorption edge compared with that of bulk g-C$_3$N$_4$, which may be caused by the quantum confinement effect due to the nanostructure properties of g-C$_3$N$_4$. According to the Kubelka–Munk method, the energy band gaps of the MCN, CMCN, and UMCN were estimated to be 2.47, 2.49, and 2.57 eV, respectively.

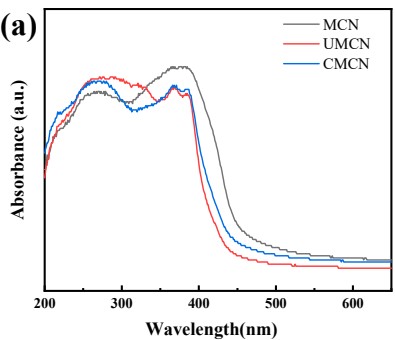 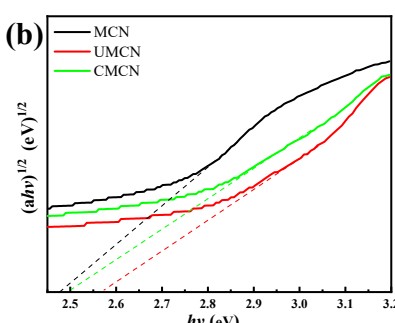

**Figure 4.** (**a**) UV–vis diffuse reflectance spectra (**b**) corresponding plots of $(\alpha h v)^{1/2}$ vs. photon energy of g-C$_3$N$_4$ prepared with different precursors.

### 2.2. Thermal Decomposition Process Analysis

Figure 5 shows the TG and DTG curves of the annealing of different precursors under the nitrogen atmosphere. Figure 5a,b shows the TG and DTG curves of MCN. There was only one obvious weight loss in the decomposition of the melamine precursor, and the weight loss increased significantly when the temperature was over 210 °C, which was due to the polycondensation of melamine to form melam and melem. For the generating of g-C$_3$N$_4$, the melamine precursor initially occurred through deamination to form melam and melem [41,42]. When the temperature was over 420 °C, the mass change rate of the DTG curves (Figure 5b) were close to zero. Figure 5c displays the TG curves of UMCN, which used a mixture of urea and melamine as precursor, with corresponding DTG curves (In Figure 5d). Figure 5c shows three obvious weight losses in the TG curve, indicating more stages of pyrolysis. Compared with MCN using the single melamine as precursor, the initial decomposition process under the temperature ranged from about 130 to 210 °C. The first weight loss could be attributed to the polycondensation of urea in the UMCN for the formation of biuret. After the first sharp peak, more weak peaks appeared. The following weight loss could be attributed to the polymerization of the forming melamine, which further formed melam and melem at about 380 °C, and then deaminated to form g-C$_3$N$_4$. During the decomposition processes, NH$_3$ or HNCO were released. The CMCN samples displayed a weak weight loss under the temperature of 225–250 °C and a sharp weight loss when the temperature was over 250 °C. In the second decomposition process, the melamine underwent thermal polymerization to form melam and melem.

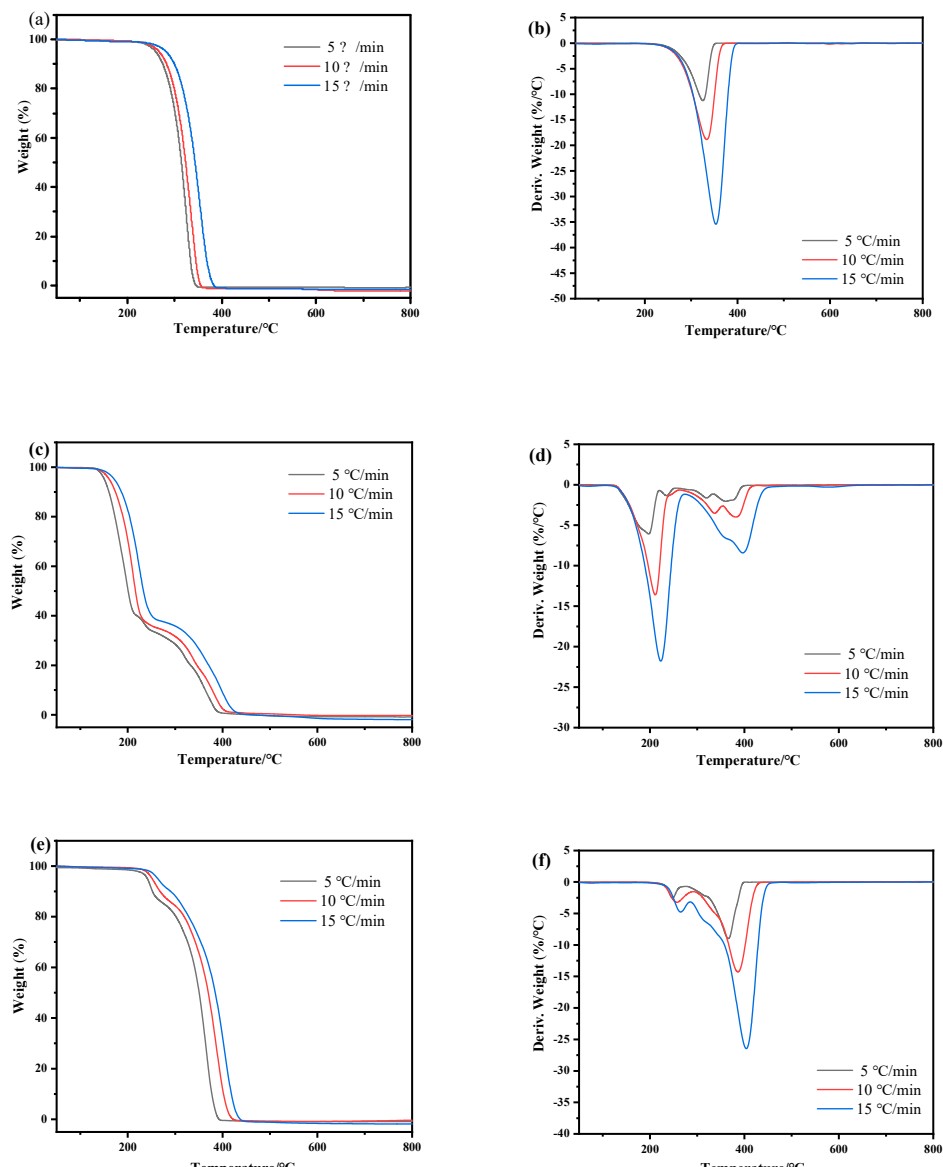

**Figure 5.** TG−DTG curves of MCN (**a**,**b**), UMCN (**c**,**d**), and CMCN (**e**,**f**).

In order to understand the relationship and interaction between the precursor and the targeted catalyst, the decomposition process of MCN, UMCN, and CMCN was analyzed using thermal decomposition kinetics. The thermal decomposition with different heating rates were measured (5 °C, 10 °C, and 20 °C) to obtain the decomposition kinetics. The model—free method without predicting the reaction mechanism was adopted due to the unknown reaction model of complex reactions. Therefore, the Flynn–Wall–Ozawa method (FWO) was used to summarize the thermal decomposition characteristics. The FWO method was used to calculate the activation energy *E*. The formula is as follows:

$$\lg\beta = \lg(AE/RG(\alpha)) - 2.315 - 0.4567E/RT$$

where *β* is the heating rate, *R* is the gas constant, *α* is the extent of reaction, *G(α)* is an integral form of the kinetic function, and *A* is the pre-exponential factor. For different heating rates at a constant *α*, the value of activation energy *E* can be calculated from the slope of the straight line of the plot *lgβ* vs. *1/T*. The conversion rate *α* is generally 0.1, 0.2, 0.3, 0.4, 0.5, 0.6, 0.7, 0.8, 0.9. In Figure 6, the dependence of activation energy on the extent of conversion for MCN, UMCN, and CMCN using Friedman's method is displayed.

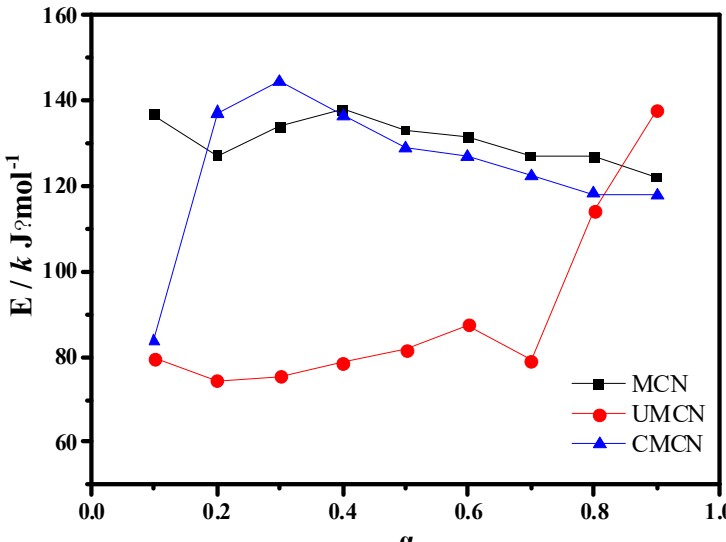

**Figure 6.** Dependence of activation energy on the conversion rate α for MCN, UMCN, and CMCN.

Activation energy *E* for thermal decomposition implies the magnitude of reaction resistance and difficulty of the decomposition process, which is increased with the increase of reaction resistance. For MCN, the activation energy changed slightly with the conversion rate, which indicated that the dissociation process could be a simple one-step process. Compared with that of MCN and CMCN, the thermal decomposition activation energy of UMCN sustained a lower level when the conversion was under 0.7. Then UMCN required higher activation energy when the conversion was higher than 0.8. The results indicate that, as for UMCN, there could be a distinct space steric effect when the conversion increased to higher than 0.8, which made the reaction resistance increase and caused the melamine to melem conversion process to be more difficult.

### 2.3. Photocatalytic Activity of Samples Prepared by Different Precursors

The UV-vis absorption spectrum and molecular formula of NOR is shown in Figure 7a, which is a combination with a benzene ring and a conjugate double bond. The absorption spectra (Figure 7a) were obtained by scanning λ between 232–358 nm. The measured sample had the maximum absorption peak at 273 nm, and the excipients had no interference at 273 nm. NOR is a third-generation fluoroquinolone antibiotic. As a colorless organic pollutant, it can exclude the photosensitization of colored dyes and further evaluate the photocatalytic properties of the material. The main structure of NOR includes the piperazine part and the quinolone part. The benzene rings in these two parts have the highest electron density value. NOR antibiotics can be adsorbed on the surface of the adsorbent by the electrostatic gravity, van der Waals force, and ion exchange of the adsorbent. At the same time, the fluorine atomic group and carboxyl group can strengthen the adsorption with the adsorbent in water.

The sorption ability for NOR in solution using MCN, UMCN, and CMCN samples in the absence of light was performed in Figure 7b, and the sorption attained desorption and absorption equilibrium within 60 min. The NOR adsorption rate was about 4%, 15%, and 14% using MCN, UMCN, and CMCN, respectively. As a control test, the photolysis effect of NOR under visible-light irradiation was investigated in Figure 7c, and the results indicate that the effect of NOR was minimal. The photocatalytic activity of the MCN, UMCN, and CMCN was evaluated by the degradation of NOR solutions under visible-light irradiation, as shown in Figure 7c. The CMCN and UMCN displayed better photocatalytic activity than MCN. The UMCN possessed outstanding photocatalytic activity for NOR degradation compared with MCN. Using UMCN as the catalyst, the NOR degradation rate attained nearly 54% within 15 min, then the degradation gradually slowed down, eventually

attaining about 92% within 150 min. The pseudo-first-order reaction kinetics model was used to express the degradation rate, and the degradation rate of NOR conformed to the pseudo-first-order reaction kinetics equation:

$$-Ln\left(\frac{C_t}{C_0}\right) = k_{obs} * T$$

where $k_{obs}$ (min$^{-1}$) is the kinetic rate constant, and the larger the $k_{obs}$ value, the faster the degradation rate; $C_0$ is the initial concentration of NOR, $T$ is the reaction time. Kinetic fitting was performed on the experimental results, and the apparent reaction rate constant $k$ was shown in Figure 7d. The obtained reaction rate constant $k$ of MCN, UMCN, CMCN was 1.19, 1.50, and 1.21 min$^{-1}$, respectively. The results indicate that UMCN catalysts possessed the best photocatalytic performance for the degradation of NOR.

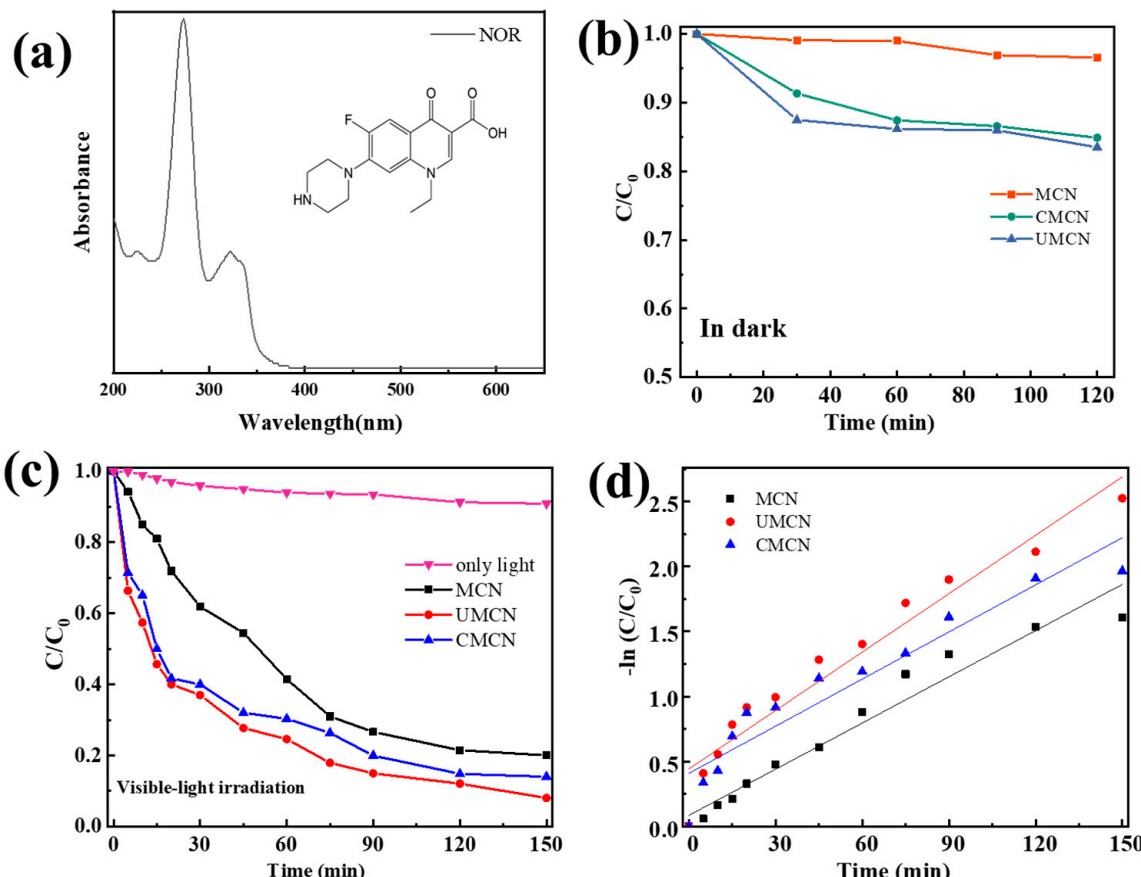

**Figure 7.** (**a**) UV-vis absorption spectrum and (inset) the molecular formula of NOR, (**b**) adsorption ability for NOR using MCN, UMCN, and CMCN samples in the dark, (**c**) photocatalytic efficiency, and (**d**) corresponding reaction kinetics as a function of time using MCN, UMCN, and CMCN samples for the removal of NOR under visible-light irradiation.

The photocatalytic performance of UMCN in the NOR solution with different initial pH was evaluated. Generally, as shown in Figure 8a, the UMCN sustained a good degradation efficiency, which indicated the stable performance of UMCN within the solution pH range from 4.5 to 9. To further test its cycling degradation performance, four-cycling degradation is shown in Figure 8b. After the fourth cycle degradation, UMCN sustained a degradation rate of NOR above 75% under visible light irradiation, indicating its potential for a long-term ability.

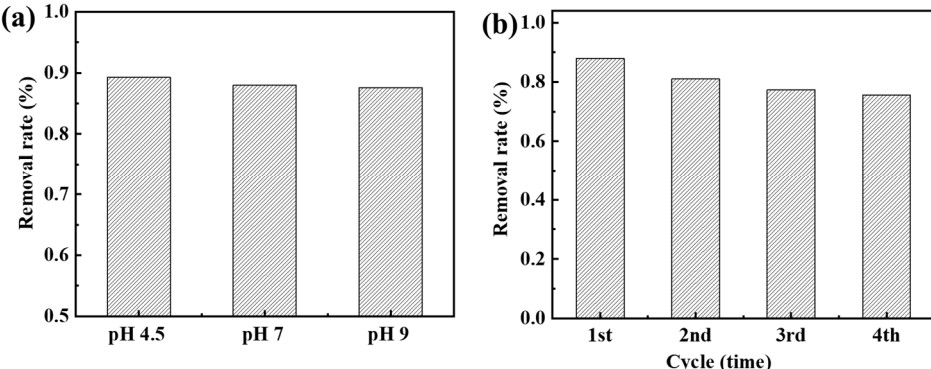

**Figure 8.** (**a**) Effects of solution pH on the NOR degradation and (**b**) cycling tests using UMCN catalysts under 120 min visible light irradiation. (Experimental conditions: NOR concentration = 10 mg/L, photocatalysts amount = 1 g/L).

Scheme 2 illustrates the photocatalytic degradation of NOR solution by g-C$_3$N$_4$ nanotubes. During the photocatalytic reaction, the visible-light excited photogenerated electrons in the CB of g-C$_3$N$_4$ reduce O$_2$ to O$_2^{-\bullet}$. In addition, $\bullet$OH could be generated via a multistep reaction. Finally, O$_2^{-\bullet}$ and $\bullet$OH as the strong oxygen free radicals facilitated the degradation of NOR. The better photocatalytic performance of UMCN and CMCN could be attributed to their more suitable microstructure. The nanotubular structures provided easier mass transfer efficiency and higher photogenerated charge mobility for the UMCN and CMCN. Furthermore, abundant amino groups on the surface of the UMCN facilitating the formation of reaction active site on the surface for the photocatalytic process.

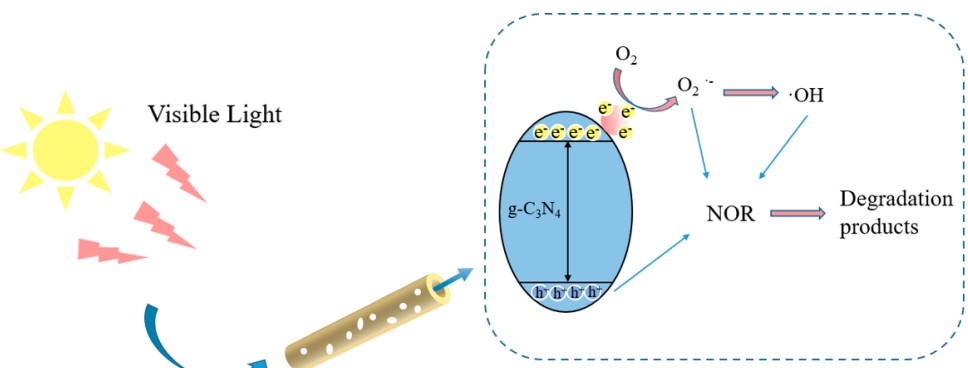

**Scheme 2.** The photocatalytic degradation of NOR solution by the g-C$_3$N$_4$ nanotubes.

### 3. Conclusions

In summary, the g-C$_3$N$_4$ was prepared by a one-pot thermal decomposition process based on controlling precursors. Melamine was used as a N-rich molecular precursor and a mixture with urea or cyanuric acid was also prepared (UMCN and CMCN). The SEM results show that, different from the sheet-like morphology obtained by single precursor melamine, the special one-dimensional nanotube structure of g-C$_3$N$_4$ (the UMCN and CMCN) was obtained by forming supramolecular polymer precursors with a mixture of melamine/urea or cyanuric acid. The exposed surfaces of nanotubes thus created more catalytic active sites for the photocatalytic degradation of NOR. In addition, compared with that of bulk g-C$_3$N$_4$, there more amino groups were retained on the surface of UMCN. The TG-DTG results show that there was more than one obvious weight loss in the decomposition process for UMCN and CMCN. The dissociation involved a multiple-step process. The UMCN and CMCN showed much better photocatalytic performance for the degradation of NOR antibiotics in solution than MCN.

## 4. Materials and Methods

### 4.1. Preparation of Catalysts

All chemicals and reagents in the experiment were commercially available and used without further purification. Deionized water was used throughout the experiment. The catalysts were synthesized through a thermal polymerization method. In brief, 5 g melamine was transferred into a 50 mL ceramic crucible with a lid, then heated to 550 °C with a heating rate of 5 °C/min and kept for four hours in a muffle furnace. The yellow residual was ground into a powder using a mortar to obtain g-$C_3N_4$, denoted as MCN. Meanwhile, different precursors were used to prepare the g-$C_3N_4$ catalysts, which were prepared with the similar method mentioned above but urea was replaced with a mixture of 5 g urea and 0.5 g melamine, a mixture of 1 g melamine and 5 g cyanuric acid, and denoted as MCN, UMCN, CMCN, respectively.

### 4.2. Photocatalytic Measurements

Norfloxacin (NOR) was used as model contaminants to investigate the photocatalytic performance of the catalysts for removing organic contaminants. The photocatalytic performance of samples was performed in a 500 mL glass beaker. A parallel photochemical reaction instrument (CEL-LAB200E7, Beijing, China) was used, with 420 nm LED light as visible light source. For each experiment, 50 mg of catalysts were dispersed in 50 mL (10 mg/L) NOR aqueous solution. Prior to visible light irradiation, the reaction solution was stirred for 60 min in the dark to reach an adsorption–desorption equilibrium. At given time intervals, about 2 mL reaction solution was taken through a 0.22 μm cellulose acetate filter, followed by measuring with a UV-vis spectrophotometer (UV-2600, Shimadzu, Kyoto, Japan) at 273 nm to detect the residual NOR concentration. Based on the Lamber–Mier law and the Beer–Lambert law, the pollutant removal rate was calculated by the formula, *Removal rate% = $(A_0 - A_t)/A_0 * 100\%$*, where the absorbance occurs at the reaction time of 0 min, and A is the instantaneous absorbance during the reaction.

### 4.3. Characterization

The crystal structures of the catalysts were examined by an X-ray diffractometer (Panalytical X'Pert'3 Powder, Almelo, The Netherlands). The morphology structure of the catalysts was analyzed by scanning electronic micrographs (SEM, Zeiss Sigma 300, Jena, Germany). Fourier transformed infrared spectra (FT-IR) was performed by a Thermo Scientific Nicolet IS20 spectrometer (Waltham, MA, USA). The optical properties of catalysts were analyzed by the UV–vis diffuse reflectance spectra (DRS) test on a spectrometer (Shimadzu UV-2600, Kyoto, Japan). The comprehensive thermal analysis was carried out by a thermogravimetric analyzer (NETZSCH STA 449F3, Bayern, Germany). The samples were placed in the corundum crucible and calcined at the heating rate of 5, 10, 20 °C/min.

**Author Contributions:** Conceptualization, H.G.; investigation, X.L., X.X. and M.Y.; writing—original draft preparation, X.L., X.X. and M.Y.; writing—review and editing, H.G. and Y.H.; project administration, H.G. and Y.H.; funding acquisition, H.G. and Y.H. All authors have read and agreed to the published version of the manuscript.

**Funding:** This research was supported by the Zhejiang Provincial Natural Science Foundation of China under grant LY21E090004, National Natural Science Foundation of China under grant 52070103, the Natural Science Foundation of Ningbo under grant 202003N4135, the General Research Project of Zhejiang Provincial Department of Education under grant Y202043966, and K.C. Wong Magna Fund in Ningbo University.

**Data Availability Statement:** Not applicable.

**Conflicts of Interest:** The authors declare no conflict of interest.

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
