# Peer review of "The Effect of Different g-C3N4 Precursor Nature on Its Structural Control and Photocatalytic Degradation Activity"

_catalysts, doi:10.3390/catal13050848_

Round 1

Reviewer 1 Report

1) Line 24-46, PP 1-2. Too much explanation on the wastewater, environmental effect and treatment used. Kindly reduce those explanation since the focus of this research is on material.

2)Line 59, pp2, no citation was included on each of CN preparation.

3) In the literature review part, kindly added type of precursors used for gCN on previous works.

4) Line 138; Kindly breaf the sample preparation name for every acronym used exp; MCN, UMCN and CMCN.

5) Figure 1 should be placed after a few explanations text in the results and discussion

6) Kindly added text explaining Fig 2 (FTIR) 

7) English must be thoroughly check. There are still a few grammatical errors.  

Author Response

The effect of different g-C3N4 precursor nature on its structural control and photocatalytic degradation activity

Manuscript ID: catalysts-2311579

Title: The effect of different g-C3N4 precursor nature on its structural control and photocatalytic degradation activity

Authors: Xiuhang Liu, Xiaoye Xu, Huihui Gan*, Mengfei Yu, Ying Huang*

Response to Reviewer #1’s comments

We are very grateful for the Reviewer#1’ constructive and insightful comments and suggestions. We have studied the comments carefully and have made corrections which we hope to meet with approval. Below are our detailed responses to the concern from Reviewer#1.

Reviewer#1

1) Line 24-46, PP 1-2. Too much explanation on the wastewater, environmental effect and treatment used. Kindly reduce those explanation since the focus of this research is on material.

Response: The explanation on the wastewater, environmental effect and treatment used has been reduced in the Introduction section. In the revised manuscript, we have made the introduction more concise and specific.

The modified part in Introduction is listed as follows.

The explanation in Line 28-30, Page 1, “Antibiotic pollutants, as refractory organic pollutants, are challenging to be removed through the traditional water treatment process in urban sewage treatment plants, so they can exist stably in the environment for a long time and eventually pass into the human body through drinking water or food chain”, has been deleted in the revised manuscript.

The explanation in Line 33-35, Page 1, “The treatment technology of antibiotics has been intensively studied, such as conventional additive coagulation treatment process, activated carbon adsorption, membrane separation technology and other deep purification technologies”, has been deleted in the revised manuscript.

  • Line 59, pp2, no citation was included on each of CN preparation.

Response: The references related to the CN preparation have been added in the Introduction section.

The added references are listed as follows.

  1. Mohazzab, B.F.; Akhundi, A.; Rahimi, K.; Jaleh, B.; Moshfegh, A.Z. P-Doped g-C3N4 Nanosheet-Modified BiVO4 Hybrid Nanostructure as an Efficient Visible Light-Driven Water Splitting Photoanode. ACS Applied Energy Materials 2022, 5, 12283-12
  2. Wan, Z.; Zhang, G.K.; Wu, X.Y.; Yin, S. Novel visible-light-driven Z-scheme Bi12GeO20/g-C3N4 photocatalyst: Oxygen-induced pathway of organic pollutants degradation and proton assisted electron transfer mechanism of Cr(VI) reduction. Applied Catalysis B: Environmental 2017, 207, 17-26.
  3. Liu, J.; Guo, H.; Yin, H.; Nie, Q.; Zou, S. Accelerated Photodegradation of Organic Polluta nts over BiOBr/Protonated g-C3N4. Catalysts 2022, 12, 1109.
  4. Ghanbari, M.; Salavati-Niasari, Copper iodide decorated graphitic carbon nitride sheets with enhanced visible-light response for photocatalytic organic pollutant removal and antibacterial activities. Ecotoxicology and Environmental Safety 2021, 208, 111712.

3) In the literature review part, kindly added type of precursors used for g-CN on previous works.

Response: The previous works related to the types of precursors used for g-C3N4 have been added in the Introduction section, in Line 68-69, Page 2.

The added content in Introduction is listed as follows:

The N-rich molecular precursors such as melamine, urea and thiourea have been widely used for the photocatalysis.

4) Line 138; Kindly breaf the sample preparation name for every acronym used exp; MCN, UMCN and CMCN.

Response: The explanation of MCN, UMCN and CMCN have been added in the revised manuscript, in Line 133-136, Page 3.

The added content in Introduction is listed as follows:

The thermal decomposition of melamine synthesized the catalyst samples, denoted as MCN, and prepared by a mixture of urea and melamine, denoted as UMCN, a mixture of melamine and cyanuric acid as precursors denoted as CMCN.

5) Figure 1 should be placed after a few explanations text in the results and discussion

Response: The place of Figure 1 in the Results and Discussion has been adjusted and put after a few explanations text, in Line 149, Page 4.

6) Kindly added text explaining Fig 2 (FTIR)

Response: The text explaining of Figure 2 has been added in the revised manuscript, in Line 166-167, Page4.

The added content is listed as follows:

The molecular structure of MCN, UMCN and CMCN were characterized by Fourier transform infrared (FTIR) spectrometry, and the FTIR spectra are shown in Fig. 2.

  • English must be thoroughly check. There are still a few grammatical errors.

Response: The English has been thoroughly checked and the grammatical errors has been modified.

Reviewer 2 Report

In the manuscript entitled “The regulation of g-C3N4 structure and photocatalytic activity based on different precursors controlling” with Reference Number “catalysts-2311579”, the authors investigate the relationship between precursor types on the microstructure and photocatalytic activity of g-C3N4 in the photocatalytic degradation of norfloxacin. The photocatalyst was sufficiently characterized by XRD, FT-IR, UV-Vis-DRS, SEM. Especially, the optimum photocatalyst shows advantages of enhanced photocatalytic efficiency. So, I think this manuscript has a potential to be accepted for publication after implementation of comments and improvements. My detailed comments are as follows:

Title

The title is not very clear straight forward.  I suggest the following title

“The effect of different g-C3N4 precursor nature on its structural control and   photocatalytic degradation activity”

Abstract

-1-Please include some important quantitative results in the abstract.

 Introduction

2-References 13-15 are old.  In addition, they must be specified for each subject area in order to be useful for readers

Please update the references and prepare as following:

 Water splitting (ACS Applied Energy Materials 5 (10), 2022, 12283-12296),

Pollutant degradation (Catalysts 2022, 12, 1109-1118)

Carbon dioxide capture (Journal of Environ. Chem. Eng., 9, 2021,104631)

 3-Tuning porous and hollow microstructure, such as nanosheets, nanoflower and nanorod of g-C3N4 [25-28].  This statement is not useful for readers. Because a reader must open all four articles to find his/her desire findings.  Please update the references and prepare like the following:

Nanosheets (RSC Adv.,7, 2017, 2333-2341)

Nanoflowers (Journal of Environmental management 274 2020, 111208).

4-Materials and methods must be moved before Results and Discussion section.

 Results and Discussion

5-The wide-angle XRD patterns of MCN, UMCN and CMCN is provided in Figure 1. In order to investigate the porous structure of the UMCN and CMCN, small angle region of XRD pattern could to be provided.

6-There are many porous g-CN reported in literatures; the authors should summarize and compare what they prepared with the reported ones.

 7-The surface area of MCN, UMCN and CMCN should be reported. Then, the effect of the porous should be discussed.

8-The effect of reaction conditions such as photocatalyst dosage, NOR concentration and pH should be investigated.

9-Please add the result of “Photolysis” in Figure 7b as a reference point for your three prepared photocatalysts.

 10-In order to examine the photocatalytic activity and determine dominant reactive species in the reaction, scavenger tests should be examined.

11-The stability and recyclability of the photocatalyst are the most important aspects in the green synthesis.  Pleases provide data and discuss the issue.

 12-The manuscript does not contain Conclusion section.  Please provide conclusion.

 References:

13-Format for writing some of references are different with others such as 1, 3, 4, 19, 23.  Please make them in a similar format with the “Catalyst” Journal standard.

 14-What is the page number for Ref. 2?

 15-What is the volume number for Ref. 26?

 16-There are some grammar mistakes throughout the manuscript. Please recheck the manuscript for grammar and typographical errors.  See for example below:

 i) In the abstract, “UV–vis diffuse reflectance spectra” should be “UV–vis diffuse reflectance spectroscopy” to be consistent with other techniques such as microscopy.

 ii) In the abstract, “fourier transform infrared spectrometry” should be “Fourier transform infrared spectroscopy”

 iii) Page 8, line 245 – “Activation energy E, as the magnitude of reaction resistance, increases with the increase of reaction resistance. This statement is not clear.

Author Response

The effect of different g-C3N4 precursor nature on its structural control and photocatalytic degradation activity

Manuscript ID: catalysts-2311579

Title: The effect of different g-C3N4 precursor nature on its structural control and photocatalytic degradation activity

Authors: Xiuhang Liu, Xiaoye Xu, Huihui Gan*, Mengfei Yu, Ying Huang*

Response to Reviewer #2’s comments

We are very grateful for the Reviewer#2’ constructive and insightful comments and suggestions. We have studied the comments carefully and have made corrections which we hope to meet with approval. Below are our detailed responses to the concern from Reviewer#2.

Title

The title is not very clear straight forward. I suggest the following title “The effect of different g-C3N4 precursor nature on its structural control and photocatalytic degradation activity”

Response: Thank you for your suggestion. The title has been replaced by “The effect of different g-C3N4 precursor nature on its structural control and photocatalytic degradation activity”.

Abstract

1-Please include some important quantitative results in the abstract.

Response: The BET results have been added in the Abstract. in Line 16-18, Page1.

The added quantitative results are as follows:

The maximum BET surface area was 98.04 cm3g-1 through precursor controlling, more than 8 times than that of MCN (11.15 cm3g-1) using melamine as precursor.

Introduction

2-References 13-15 are old. In addition, they must be specified for each subject area in order to be useful for readers.

Please update the references and prepare as following:

Water splitting (ACS Applied Energy Materials 5 (10), 2022, 12283-12296)

Pollutant degradation (Catalysts 2022, 12, 1109-1118)

Carbon dioxide capture (Journal of Environ. Chem. Eng., 9, 2021,104631)

Response: The related references have been updated and specified for the subject area. in the Introduction section.

The updated reference are as follows:

12.Mohazzab, B.F.; Akhundi, A.; Rahimi, K.; Jaleh, B.; Moshfegh, A.Z. P-Doped g-C3N4 Nanosheet-Modified BiVO4 Hybrid Nanostructure as an Efficient Visible Light-Driven Water Splitting Photoanode. ACS Applied Energy Materials 2022, 5, 12283-12296.

13.Wan, Z.; Zhang, G.K.; Wu, X.Y.; Yin, S. Novel visible-light-driven Z-scheme Bi12GeO20/g-C3N4 photocatalyst: Oxygen-induced pathway of organic pollutants degradation and proton assisted electron transfer mechanism of Cr(VI) reduction. Applied Catalysis B: Environmental 2017, 207, 17-26.

14.Liu, J.; Guo, H.; Yin, H.; Nie, Q.; Zou, S. Accelerated Photodegradation of Organic Pollutants over BiOBr/Protonated g-C3N4. Catalysts 2022, 12, 1109.

15.Ghanbari, M.; Salavati-Niasari, M. Copper iodide decorated graphitic carbon nitride sheets with enhanced visible-light response for photocatalytic organic pollutant removal and antibacterial activities. Ecotoxicology and Environmental Safety 2021, 208, 111712.

3-Tuning porous and hollow microstructure, such as nanosheets, nanoflower and nanorod of g-C3N4 [25-28]. This statement is not useful for readers. Because a reader must open all four articles to find his/her desire findings. Please update the references and prepare like the following:

Nanosheets (RSC Adv.,7, 2017, 2333-2341)

Nanoflowers (Journal of Environmental management 274, 2020, 111208).

Response: The old references 26-28 have been updated and the statement has been modified in the revised manuscript (in Line 102-103, Page 3).

The updated content and added reference are as follows:

Self-templating of melamine with urea, cyanuric acid and cyanuric chloride, is a simple, safe and cheap method to tuning porous and hollow microstructure, such as nanorods [25], nanosheets [26], nanoflower [27]and nanotubes [28] of g-C3N4.

The updated reference are as follows:

  1. Yang, Y.; Chen, J.; Mao, Z.; An, N.; Wang, D.; Fahlman, B. D. Ultrathin g-C3N4 nanosheets with an extended visible-light-responsive range for significant enhancement of photocatalysis. RSC advances, 2017, 7, 2333-2341.
  2. Monga, D.; Ilager, D.; Shetti, N. P.; Basu, S.; Aminabhavi, T. M. 2D/2d heterojunction of MoS2/g-C3N4 nanoflowers for enhanced visible-light-driven photocatalytic and electrochemical degradation of organic pollutants. Journal of Environmental Management, 2020, 274, 111208.
  3. Wang, Z.; Chen, M.; Yu, H.; Shi, X.; Zhang, Y. Self-Assembly Synthesis of Boron-Doped Graphitic Carbon Nitride Hollow Tubes for Enhanced Photocatalytic NOx Removal under Visible Light. Applied Catalysis B Environmental, 2018, 239, 55-59.

Results and Discussion

4-Materials and methods must be moved before Results and Discussion section.

Response: The place of each section was placed following the journal template of Catalysts. Followed the instruction, we put the Materials and Methods section after the Results and Discussion.

5-The wide-angle XRD patterns of MCN, UMCN and CMCN is provided in Figure 1. In order to investigate the porous structure of the UMCN and CMCN, small angle region of XRD pattern could to be provided.

Response: The small angle region of XRD pattern has been added in Fig. 1 in the revised manuscript (Line 149, Page 4).

The updated Fig. 1 are as follows:

Figure 1. XRD patterns of the samples MCN, UMCN and CMCN.

6-There are many porous g-CN reported in literatures; the authors should summarize and compare what they prepared with the reported ones.

Response: The comparison of the reported porous structures and corresponding preparation methods have been added in the revised manuscript (Line 191, Page 5).

The updated content is as follows:

Table 1. The comparison of the reported porous structures and corresponding preparation methods.

Porous structure

Precursor

Mehod

Ref.

Short rod-like

Dicyandiamide with

SBA-15 template

High pressure and high temperature (HPHT) process

[38]

Nanosheets

Melamine powder

and ammonium bicarbonate

Solid-state reaction

[39]

Crafted 1D mesoporous microtubes

Dicyandiamide

Liquid-liquid interfacial self-assembly strategy

[40]

Nanotube

Melamine and urea

Thermal polymerization

This work

Nanotube

Melamine and cyanuric acid

Thermal polymerization

This work

  1. Yang, Z.X.; Chu, D.L.; Jia, G.R.; Yao, M.G.; Liu, B.B. Significantly narrowed bandgap and enhanced charge separation in porous, nitrogen-vacancy red g-C3N4 for visible light photocatalytic H2 production. Applied Surface Science 2020, 504, 144407.
  2. Zhang, R.; Zhang, X.M.; Liu, S.W.; Tong, J.W.; Kong, F.; Sun N.K.; Han X.L.; Zhang, Y.L. Enhanced photocatalytic activity and optical response mechanism of porous graphitic carbon nitride (g-C3N4) nanosheets. Materials Research Bulletin 2021, 140, 111263.
  3. Liu, Q.; Chen C.C.; Yuan K.J. Sewell, Chris D.; Zhang Z.G.; Fang X.M.; Lin Z.Q. Robust route to highly porous graphitic carbon nitride microtubes with preferred adsorption ability via rational design of one-dimension supramolecular precursors for efficient photocatalytic CO2 conversion. Nano Energy 2020, 77, 105104.

7-The surface area of MCN, UMCN and CMCN should be reported. Then, the effect of the porous should be discussed.

Response: The BET surface area, pore volume and average pore size of MCN, UMCN and CMCN has been added in Table 1 and accordingly explanations in the revised manuscript (Line 192-198, Page 6).

The added Table 1 and explanations are as follows:

The BET surface area, pore volume and average pore size of MCN, CMCN and UMCN were summarized in Table 1. The BET surface area of UMCN is 98.04 m2/g, which is more than 8 times than that of CMCN and MCN. The results implied that the UMCN could possessed the best adsorption ability and more active sites for the photocatalytic degradation.

Table 1. BET surface area, pore volume and average pore size of MCN, CMCN and UMCN.

Samples

MCN

CMCN

UMCN

BET surface area (m2/g)

11.15

11.87

98.04

Total pore volume (cm3/g)

0.10

0.12

0.58

Average pore size (nm)

36.31

40.01

23.68

8-The effect of reaction conditions such as photocatalyst dosage, NOR concentration and pH should be investigated.

Response: Based on the investigations of reaction conditions on photocatalytic performance using prepared catalysts MCN, UMCN and CMCN, the selected reaction conditions were as follows: 1.0 g/L of the photocatalyst dosage, and 10 mg/L of NOR concentration. In Typical, 50 mg of catalysts were dispersed in 50 mL (10 mg/L) NOR aqueous solution. To investigate the effect of pH on the catalytic activity, the NOR solution were adjusted from 4.5 to 9.0. After 120 min visible-light irradiation, the UMCN catalysts showed good photocatalytic stability within the pH range of 4.5 to 9.0. The added Figure 8a and explanations in the revised manuscript (Line 306, Page 10).

The added content is as follows:

The photocatalytic performance of UMCN in the NOR solution with different ini-tial pH was evaluated. Generally, as shown in Fig. 8a, the UMCN sustained a good degradation efficiency, which indicated the stable performance of UMCN within the solution pH range from 4.5 to 9. To further test its cycling degradation performance, four cycling degradation was shown in Fig. 8b. After fourth cycle degradation, UMCN sustained degradation rate of NOR above 75% under visible light irradiation, indicat-ing its potential for the long-term ability.

Figure 8. Effects of solution pH on the NOR degradation and Cycling tests using UMCN catalysts under 120 min visible light irradiation. (Experimental conditions: NOR concentration = 10 mg/L, photocatalysts amount = 1 g/L)

9-Please add the result of “Photolysis” in Figure 7b as a reference point for your three prepared photocatalysts.

Response: The photolysis effect for the NOR solution under visible-light irradiation has been added in the Figure 7c (Line 294, Page 9), and results indicated this effect of NOR was little.

10- In order to examine the photocatalytic activity and determine dominant reactive species in the reaction, scavenger tests should be examined.

Response: We investigated the effect of active species, O2•-, •OH and h+, using BQ, TBA and EDTA-2Na as a quencher, respectively. However, the result showed there were small distinctions for the inhibition of the NOR degradation using these quenchers. We will try to figure it out in the further research.

11-The stability and recyclability of the photocatalyst are the most important aspects in the green synthesis. Pleases provide data and discuss the issue.

Response: The stability and recyclability of the photocatalyst has been evaluated and the results have been added in the revised manuscript(Line299-309,Page 9-10).

The added Figure 8 and explanations are as follows:

The photocatalytic performance of UMCN in the NOR solution with different initial pH was evaluated. Generally, as shown in Fig. 8a, the UMCN sustained a good degradation efficiency, which indicated the stable performance of UMCN within the solution pH range from 4.5 to 9. To further test its cycling degradation performance, four cycling degradation was shown in Fig. 8b. After fourth cycle degradation, UMCN sustained degradation rate of NOR above 75% under visible light irradiation, indicating its potential for the long-term ability.

Figure 8. Effects of solution pH on the NOR degradation and Cycling tests using UMCN catalysts under 120 min visible light irradiation. (Experimental conditions: NOR concentration = 10 mg/L, photocatalysts amount = 1 g/L)

12-The manuscript does not contain Conclusion section.  Please provide conclusion.

Response: The Conclusion section has been added in the revised manuscript.

13-Format for writing some of references are different with others such as 1, 3, 4, 19, 23. Please make them in a similar format with the “Catalyst” Journal standard.

Response: The format for references has been checked all through according to the Journal standard.

14-What is the page number for Ref. 2?

Response: The page number has been added for Ref. 2.

15-What is the volume number for Ref. 26?

Response: The Ref. 26 has been updated.

  1. Yang, Y.; Chen, J.; Mao, Z.; An, N.; Wang, D.; Fahlman, B. D. Ultrathin g-C3N4 nanosheets with an extended visible-light-responsive range for significant enhancement of photocatalysis. RSC advances, 2017, 7, 2333-2341.

16-There are some grammar mistakes throughout the manuscript. Please recheck the manuscript for grammar and typographical errors. See for example below:

  1. i) In the abstract, “UV–vis diffuse reflectance spectra” should be “UV–vis diffuse reflectance spectroscopy” to be consistent with other techniques such as microscopy.
  2. ii) In the abstract, “fourier transform infrared spectrometry” should be “Fourier transform infrared spectroscopy”

iii) Page 8, line 245 – “Activation energy E, as the magnitude of reaction resistance, increases with the increase of reaction resistance. This statement is not clear.

Response: The English has been thoroughly checked and the grammatical and typographical errors has been modified.

Reviewer 3 Report

The article cannot be published in this form as it contains many logical and scientific contradictions

1. The requirements of the article design template were not met in terms of a separate subtitle "Conclusions" and the correct display of equations in the text of the manuscript (lines: 239, 285, 337). It makes it difficult to read.

2.Explanation of abbreviations (MCN, UMCN, CMCN) is given only at the end of the article. This makes it very difficult to understand the results.

3. Adding abbreviations to the text of the manuscript is intended to reduce the number of words in the article. Why introduce abbreviations and then use them many times in an article along with a non-abbreviated term? For example:

Norfloxacin (NOR) line 328, 116, 262, 312

Flynn-Wall-Ozawa method (FWO) line: 237, 237

4. One gets the impression that the authors absolutely do not understand the method of setting up a photocatalytic experiment. This causes a lot of controversy and questions from the reader:

4.1. The authors indicate that they illuminated the photocatalytic solution with an LED lamp (Line: 330) indicating the lamp model PLS-SXE300. However, this lamp model is a xenon source of radiation.

https://www.perfectlight.com.cn/Product/detail/id/16.html

https://www.instrumentstrade.com/pls-sxe300-300uv-xenon-light-source_p8871.html

Discharge (xenon) and diode lamps have completely different physical principles for emitting light quanta. Accordingly, different emission spectra. This misleads the reader.

4.2. It is generally accepted that hydroxide radicals are formed after the interaction of photoholes with hydroxyl ions.

https://doi.org/10.1016/j.psep.2023.01.033   Fig. 9.

https://doi.org/10.1016/j.chemosphere.2022.137654  Fig. 11-13.

https://doi.org/10.1016/j.mtsust.2022.100264 Fig. 3.

However, in Scheme 2 and in the description, the hydroxide radical is formed from the superoxide anion. It is necessary to indicate in the text (line 292-293) of the article references to works that refute the generally accepted explanations for the generation of hydroxide radicals.

4.3. In Figure 7b, there are no data on the self-decomposition of norfloxacin without a photocatalyst (only under the action of light). This does not allow an objective assessment of the photocatalytic activity of the developed materials, as well as a comparison of their efficiency.

4.4. Stopping the photocatalytic experiment at 60 minutes is completely unjustified in the text. According to Figure 7b, it is not at all clear which of the photocatalysts is more efficient: MCN - linearly reduces the concentration of the antibiotic. UMCN, CMCN after 10-20 minutes effective photocatalysis was stopped at 0.2.

4.5. The rate constant is calculated on the linear section of the kinetic curve. For MCN you can understand how the constant was calculated. However, for UMCN, CMCN the constant was calculated on which of the linear sections (by two points up to 5 minutes of reaction or by linear sections after 5 minutes of reaction)? Instead of the well-known data in Figure 7 a, it is necessary to add kinetic curves in logarithmic coordinates from which it will become clear how the reaction rate constants were determined.

4.6. The article completely lacks studies of the surface area of the obtained materials. The absence of these data does not allow an objective assessment of the results shown in Figure 7 b for samples UMCN, CMCN. According to the methodology for setting up experiments (line: 331-332), photocatalyst concentrations = 50 mg per 50 ml or 1 g/l or 1000 mg/l were used. At the same time, the antibiotic concentration was 10 mg/l. The most simplified calculation shows that if the photocatalyst has sorption activity, then it is able to absorb the antibiotic 100 times without the participation of photocatalytic mechanisms. The reviewer believes that it was the sorption activity of the photocatalyst that caused such a sharp decrease in the antibiotic concentration in Figure 7b for samples UMCN, CMCN within 5 minutes of the experiment. In this case, the photocatalytic activity of these samples is minimal and is traced by a slow linear decrease in concentration from 5 to 60 minutes of the experiment. This contradicts the conclusions about the best photoactivity of these samples. It is necessary either to show the dark (sorption) stage, or indicate the specific surface of the catalysts, or clearly explain the reasons for the sharp drop in the antibiotic concentration in Fig. 7b.

4.7. Figure 7b clearly shows that the photocatalytic process for samples UMCN, CMCN almost stopped. The authors do not clearly explain the reasons for this fact, which causes a variety of interpretations in the reader. The reviewer suggests two options:

  - up to 5 minutes - sorption activity, after 5 minutes - self-decomposition of the antibiotic under the action of a xenon lamp. Conclusion: Samples UMCN, CMCN are not photoactive!

- up to 5 minutes - photocatalysis, after 5 minutes - destruction of the catalyst structure and loss of photocatalytic activity and further self-decomposition of the antibiotic under the action of a xenon lamp. Conclusion: The photocatalyst is extremely unstable as it can work for only 5 minutes!

Why control the structure and use different precursors if the result is an unstable photocatalyst?

Cyclic testing of the obtained materials is necessary. These experiments will remove many questions about the appropriateness of the research conducted in the article.

5. If we linearly approximate the results given by the authors in Figure 7 b, then it is clearly seen that the sample MCN has the best photoactivity. This fundamentally contradicts the conclusions (line: 311-313).

Author Response

The effect of different g-C3N4 precursor nature on its structural control and photocatalytic degradation activity

Manuscript ID: catalysts-2311579

Title: The effect of different g-C3N4 precursor nature on its structural control and photocatalytic degradation activity

Authors: Xiuhang Liu, Xiaoye Xu, Huihui Gan*, Mengfei Yu, Ying Huang*

Response to Reviewer #3’s comments

We are very grateful for the Reviewer#3’ constructive and insightful comments and suggestions. We have studied the comments carefully and have made corrections which we hope to meet with approval. Below are our detailed responses to the concern from Reviewer#3.

  1. The requirements of the article design template were not met in terms of a separate subtitle "Conclusions" and the correct display of equations in the text of the manuscript (lines: 239, 285, 337). It makes it difficult to read.

Response: According to the article template of Catalysts, the Conclusions section is not mandatory. The Conclusions section has been added in the revised manuscript. The format of equations in the text also has been corrected in the revised manuscript.

  1. Explanation of abbreviations (MCN, UMCN, CMCN) is given only at the end of the article. This makes it very difficult to understand the results.

Response: The explanation of MCN, UMCN and CMCN have been added in the very beginning of Results and Discussion section, in Line133-136, Page 3.

The added content in Introduction is listed as follows:

The thermal decomposition of melamine synthesized the g-C3N4 catalyst samples, denoted as MCN, and prepared by a mixture of urea and melamine, denoted as UMCN, a mixture of melamine and cyanuric acid as precursors denoted as CMCN.

  1. Adding abbreviations to the text of the manuscript is intended to reduce the number of words in the article. Why introduce abbreviations and then use them many times in an article along with a non-abbreviated term? For example:

Norfloxacin (NOR) line 328, 116, 262, 312

Flynn-Wall-Ozawa method (FWO) line: 237, 237

Response: The formats of abbreviations and non-abbreviated term have been checked and modified in the revised manuscript.

  1. One gets the impression that the authors absolutely do not understand the method of setting up a photocatalytic experiment. This causes a lot of controversy and questions from the reader:

4.1. The authors indicate that they illuminated the photocatalytic solution with an LED lamp (Line: 330) indicating the lamp model PLS-SXE300. However, this lamp model is a xenon source of radiation.

https://www.perfectlight.com.cn/Product/detail/id/16.html

https://www.instrumentstrade.com/pls-sxe300-300uv-xenon-light-source_p8871.html

Discharge (xenon) and diode lamps have completely different physical principles for emitting light quanta. Accordingly, different emission spectra. This misleads the reader.

Response: We used the CEL-LAB200E7 parallel photochemical reaction instrument throughout the photocatalytic degradation experiment, which places the 420 nm LED light source in the center of the 10-bit reactor and rotates the LED light source. We have corrected the lamp model in the revised manuscript (Line 352, Page 11).

4.2. It is generally accepted that hydroxide radicals are formed after the interaction of photoholes with hydroxyl ions.

https://doi.org/10.1016/j.psep.2023.01.033   Fig. 9.

https://doi.org/10.1016/j.chemosphere.2022.137654  Fig. 11-13.

https://doi.org/10.1016/j.mtsust.2022.100264 Fig. 3.

However, in Scheme 2 and in the description, the hydroxide radical is formed from the superoxide anion. It is necessary to indicate in the text (line 292-293) of the article references to works that refute the generally accepted explanations for the generation of hydroxide radicals.

Response: There has been reported that •OH may be generated at the CB through •O2, as well as by direct oxidation with h+ at the VB. When it comes to g-C3N4, the VB position (VB) is less positive than that of the OH-/•OH couple, which implies the holes in the VB of g-C3N4 have weak oxidation abilities for the direct oxidation OH- or H2O to form •OH. (Phys. Chem. Chem. Phys. 15 (39) (2013) 16883–16890; Chem.–Eur. J.,2009, 15, 6731)

4.3. In Figure 7b, there are no data on the self-decomposition of norfloxacin without a photocatalyst (only under the action of light). This does not allow an objective assessment of the photocatalytic activity of the developed materials, as well as a comparison of their efficiency.

Response: The photolysis effect for the NOR solution under visible-light irradiation has been added in the Figure 7c. The photolysis effect of NOR in solution was little as shown in Fig. 7c.in Line 294, Page 9.

4.4. Stopping the photocatalytic experiment at 60 minutes is completely unjustified in the text. According to Figure 7b, it is not at all clear which of the photocatalysts is more efficient: MCN - linearly reduces the concentration of the antibiotic. UMCN, CMCN after 10-20 minutes effective photocatalysis was stopped at 0.2.

Response: Appreciation for your suggestion. We have lengthened the photocatalytic degradation time to 150 min as shown in Fig, 7c. The results showed that UMCN and CMCN possessed better photocatalytic activity than MCN within 150 min irradiation.

4.5. The rate constant is calculated on the linear section of the kinetic curve. For MCN you can understand how the constant was calculated. However, for UMCN, CMCN the constant was calculated on which of the linear sections (by two points up to 5 minutes of reaction or by linear sections after 5 minutes of reaction)? Instead of the well-known data in Figure 7 a, it is necessary to add kinetic curves in logarithmic coordinates from which it will become clear how the reaction rate constants were determined.

Response: Appreciation for your suggestion. We have lengthened the photocatalytic degradation time to 150 min as shown in Fig, 7c, and corresponding reaction kinetics has been shown in Fig. 7d.in Line 294, Page 9.

4.6. The article completely lacks studies of the surface area of the obtained materials. The absence of these data does not allow an objective assessment of the results shown in Figure 7 b for samples UMCN, CMCN. According to the methodology for setting up experiments (line: 331-332), photocatalyst concentrations = 50 mg per 50 ml or 1 g/l or 1000 mg/l were used. At the same time, the antibiotic concentration was 10 mg/l. The most simplified calculation shows that if the photocatalyst has sorption activity, then it is able to absorb the antibiotic 100 times without the participation of photocatalytic mechanisms. The reviewer believes that it was the sorption activity of the photocatalyst that caused such a sharp decrease in the antibiotic concentration in Figure 7b for samples UMCN, CMCN within 5 minutes of the experiment. In this case, the photocatalytic activity of these samples is minimal and is traced by a slow linear decrease in concentration from 5 to 60 minutes of the experiment. This contradicts the conclusions about the best photoactivity of these samples. It is necessary either to show the dark (sorption) stage, or indicate the specific surface of the catalysts, or clearly explain the reasons for the sharp drop in the antibiotic concentration in Fig. 7b.

Response: The BET surface area, pore volume and average pore size of MCN, UMCN and CMCN has been added in Table 1 and accordingly explanations in the revised manuscript (Line 192 - 198, Page 5).

The added Table 1 and explanations are as follows:

The BET surface area, pore volume and average pore size of MCN, CMCN and UMCN were summarized in Table 1. The BET surface area of UMCN is 98.04 m2/g, which is more than 8 times than that of CMCN and MCN. The results implied that the UMCN could possessed the best adsorption ability and more active sites for the photocatalytic degradation.

Table 1. BET surface area, pore volume and average pore size of MCN, CMCN and UMCN.

Samples

MCN

CMCN

UMCN

BET surface area (m2/g)

11.15

11.87

98.04

Total pore volume (cm3/g)

0.10

0.12

0.58

Average pore size (nm)

36.31

40.01

23.68

The sorption ability for NOR in solution using MCN, UMCN and CMCN samples in the absence of light was performed in Fig. 7b, and the sorption attained desorption and absorption equilibrium within 60 min. The NOR adsorption rate was about 4%, 15% and 14% using MCN, UMCN and CMCN, respectively. As a control test, the photolysis effect of NOR under visible-light irradiation was investigated, and results indicated this effect of NOR was little.

Figure 7. (a) UV-vis absorption spectrum and (inset) the molecular formula of NOR, (b) Adsorption ability for NOR using MCN, UMCN and CMCN samples in dark, (c) Photocatalytic efficiency, and (d) corresponding reaction kinetics as a function of time using MCN, UMCN and CMCN samples for the removal of NOR under visible-light irradiation.

4.7. Figure 7b clearly shows that the photocatalytic process for samples UMCN, CMCN almost stopped. The authors do not clearly explain the reasons for this fact, which causes a variety of interpretations in the reader. The reviewer suggests two options:

 - up to 5 minutes - sorption activity, after 5 minutes - self-decomposition of the antibiotic under the action of a xenon lamp. Conclusion: Samples UMCN, CMCN are not photoactive!

- up to 5 minutes - photocatalysis, after 5 minutes - destruction of the catalyst structure and loss of photocatalytic activity and further self-decomposition of the antibiotic under the action of a xenon lamp. Conclusion: The photocatalyst is extremely unstable as it can work for only 5 minutes!

Response: The self-photolysis decomposition of norfloxacin in water under the 150 min irradiation has been performed in Fig. 7c. and this decomposition performance was little.

The sorption ability for NOR in solution using MCN, UMCN and CMCN samples in the absence of light was performed in Fig. 7b, and the sorption attained desorption and absorption equilibrium within 60 min. The NOR adsorption rate was about 4%, 15% and 14% using MCN, UMCN and CMCN, respectively. The photocatalytic activity of the MCN, UMCN and CMCN was evaluated by the degradation of NOR solutions under visible-light irradiation, as shown in Fig. 7c. The CMCN and UMCN displayed better photocatalytic activity than MCN. The UMCN possessed outstanding photocatalytic activity for NOR degradation compared with MCN. Using UMCN as the catalyst, the NOR degradation rate attained nearly 54% within 15 min, then the degradation gradually slowdown, eventually attaining about 92% within 150 min. The pseudo-first-order reaction kinetics model was used to express the degradation rate. Kinetic fitting was performed on the experimental results, and the apparent reaction rate constant k was shown in Figure 7d. The obtained reaction rate constant k of MCN, UMCN, CMCN was 1.19, 1.50 and 1.21 min-1, respectively. The results indicated that UMCN catalysts possessed the best photocatalytic performance for the degradation of NOR.

Why control the structure and use different precursors if the result is an unstable photocatalyst?

Cyclic testing of the obtained materials is necessary. These experiments will remove many questions about the appropriateness of the research conducted in the article.

Response: The Cyclic testing of the obtained materials was added in the revised manuscript (Line 294, Page 9), the results indicated that after fourth cycle degradation, UMCN sustained degradation rate of NOR above 75% under visible light irradiation, indicating its potential for the long-term ability. In addition, within the solution pH range from 4.5 to 9, UMCN maintained the stable degradation performance.

The added Figure 8b and explanations are as follows:

The photocatalytic performance of UMCN in the NOR solution with different initial pH was evaluated. Generally, as shown in Fig. 8a, the UMCN sustained a good degradation efficiency, which indicated the stable performance of UMCN within the solution pH range from 4.5 to 9. To further test its cycling degradation performance, four cycling degradation was shown in Fig. 8b. After fourth cycle degradation, UMCN sustained degradation rate of NOR above 75% under visible light irradiation, indicating its potential for the long-term ability.

Figure 8. Effects of solution pH on the NOR degradation and Cycling tests using UMCN catalysts under 120 min visible light irradiation. (Experimental conditions: NOR concentration = 10 mg/L, photocatalysts amount = 1 g/L)

  1. If we linearly approximate the results given by the authors in Figure 7 b, then it is clearly seen that the sample MCN has the best photoactivity. This fundamentally contradicts the conclusions (line: 311-313).

Response: The photocatalytic process could not be described and predicted very well using the pseudo first-order kinetic when it had been going on for 5 minutes to 60 min. Following the reviewers' suggestion, we have lengthened the photocatalytic degradation time to 150 min. The results indicated that UMCN still exhibited the best photocatalytic performance for NOR degradation.

Round 2

Reviewer 3 Report

The authors have substantially revised the article. In its current form, the article has a logically structured text that is understandable to the reader. The article may be published in its current form.